# Multifaceted Interplay between Hormones, Growth Factors and Hypoxia in the Tumor Microenvironment

**DOI:** 10.3390/cancers14030539

**Published:** 2022-01-21

**Authors:** Rosamaria Lappano, Lauren A. Todd, Mia Stanic, Qi Cai, Marcello Maggiolini, Francesco Marincola, Violena Pietrobon

**Affiliations:** 1Department of Pharmacy, Health and Nutritional Sciences, University of Calabria, 87036 Rende, Italy; marcello.maggiolini@unical.it; 2Department of Biology, University of Waterloo, Waterloo, ON N2L 3G1, Canada; lauren.todd@uwaterloo.ca; 3Department of Laboratory Medicine & Pathobiology, University of Toronto, Toronto, ON M5S 1A8, Canada; mia.stanic311@gmail.com; 4Kite Pharma Inc., Santa Monica, CA 90404, USA; qi.cai2011@gmail.com (Q.C.); fmarincola@gmail.com (F.M.)

**Keywords:** growth factors, hormones, hypoxia-inducible factors, hypoxia, immune exclusion, immunotherapies, cancer

## Abstract

**Simple Summary:**

Hormones and growth factors impact many processes in the cell. Moreover, these molecules influence tumor growth, as does a lack of oxygen (hypoxia) that characterizes cancer progression. Proteins that are stabilized by low oxygen tension, known as hypoxia-inducible factors (HIFs), help tumor cells to adapt to their environment. Of note, hormones and growth factors regulate the activity of HIFs toward malignant aggressiveness, including the resistance to therapy. In this review, we summarize the current knowledge regarding the role of hormones and growth factors in cancer development with a particular focus on their interplay with hypoxia and HIFs and comment on how these factors influence the response to cancer immunotherapy.

**Abstract:**

Hormones and growth factors (GFs) are signaling molecules implicated in the regulation of a variety of cellular processes. They play important roles in both healthy and tumor cells, where they function by binding to specific receptors on target cells and activating downstream signaling cascades. The stages of tumor progression are influenced by hormones and GF signaling. Hypoxia, a hallmark of cancer progression, contributes to tumor plasticity and heterogeneity. Most solid tumors contain a hypoxic core due to rapid cellular proliferation that outgrows the blood supply. In these circumstances, hypoxia-inducible factors (HIFs) play a central role in the adaptation of tumor cells to their new environment, dramatically reshaping their transcriptional profile. HIF signaling is modulated by a variety of factors including hormones and GFs, which activate signaling pathways that enhance tumor growth and metastatic potential and impair responses to therapy. In this review, we summarize the role of hormones and GFs during cancer onset and progression with a particular focus on hypoxia and the interplay with HIF proteins. We also discuss how hypoxia influences the efficacy of cancer immunotherapy, considering that a hypoxic environment may act as a determinant of the immune-excluded phenotype and a major hindrance to the success of adoptive cell therapies.

## 1. Importance of Hormones and Growth Factors (GFs) in Tumor Onset and Progression

Hormones and GFs are described as endogenous chemical components that act as cellular signaling factors to regulate a variety of processes such as cell growth, maturation, differentiation, function, and metabolism [1,2]. They exert their functions by binding to specific receptors on target cells to activate downstream signaling cascades. This activation can suppress or enhance the aforementioned processes. Such cellular responses are frequently described as negative or positive feedback loops [2].

Hormones and GFs are essential to the regulation of crucial processes in healthy cells. However, they also significantly affect metabolic processes in tumor cells. Moreover, despite having a direct effect on single tumor cells, hormones and GFs can also mediate the interplay between tumor cells, their interaction with the extracellular matrix (ECM), as well as their interaction with cells of surrounding tissues [3]. Such complex interactions significantly affect processes that influence tumor behavior such as proliferation, angiogenesis, or local inflammatory responses [4,5].

Many tumor cells frequently do not depend on the synthesis of signaling molecules from neighboring cells but can synthetize them independently and thus mediate their own metabolism and proliferation [6]. The secretion of hormones and GFs by cancer cells is also known as ectopic secretion and is often associated with tumors of endocrine tissues such as pancreatic islet cells, parathyroid cells, thyroid cells, or neuroendocrine cells that can be found in nearly every organ [7,8]. Hormones and GFs that influence tumor growth and progression can be classified into several groups, amongst the most important being steroid hormones such as estrogens and androgens, Epidermal GFs (EGFs) and Neuregulins (NRGs), Insulin-like GFs (IGFs), Transforming GF-β (TGF-β), and Vascular Endothelial GFs (VEGFs) [5].

Thus, hormones and GFs can affect various aspects of tumor cell metabolism. However, considering that tumors can arise from various cell and tissue types, different tumors can activate different molecular players and mechanisms to facilitate their growth and progression. The following sections will provide an overview of the key hormones and GFs, pathways, and mechanisms driving tumor cell metabolism.

### 1.1. Hormone and GF-Mediated Regulation of Intracellular Signaling Cascades

Hormones that affect the growth of tumor cells are usually lipid-soluble molecules that can pass directly through the cellular membrane and bind to intracellular proteins or nuclear receptors to mediate downstream signaling events and gene expression. In contrast, most GFs require engagement with a specific cell-surface receptor to enter the cell and mediate cellular signaling [9]. Accordingly, hormones mainly influence the expression of target genes, whereas GFs can directly mediate processes that take place in the cytoplasm. Moreover, GFs can also influence gene expression. Despite activating classical signaling cascades, GFs can indirectly or directly function as transcription factors, as well as induce the activity of transcription factors, like Signal Transducers and Activators of Transcription (STATs), or SMADs, and thus mediate gene expression [5,10,11,12,13]. Thus, both hormones and GFs ultimately mediate important cellular processes [14]. For example, upon GF-receptor engagement or the activation of hormone-responsive genes, protein phosphorylation events and protein kinase activities are crucial in the transmission of growth signals [9,14]. Prominent signaling cascades that are subsequently activated include various mitogenic pathways such as the MAPK/ERK, SMAD, PI3K/AKT, Ras-like GTPases, or the phospholipase C-γ pathways [9].

Overall, ligand-receptor binding is a crucial step in the activation of molecular signaling cascades. A prominent example of ligand-receptor binding that influences cellular growth is the activation of the Growth Hormone Receptor (GHR) upon Growth Hormone (GH) binding. Their engagement results in phosphorylation of the receptor-associated kinase Janus Kinase-2 (JAK2) and activation of STAT5, which are known mediators of GH function and oncogenesis [13,15]. GH has also been demonstrated to affect the tissue-specific expression of microRNAs, indicating a novel role for GH beyond its classical function in the activation of signaling pathways [13,16]. Another example of GF binding that affects downstream cellular signaling pathways that underlie the development of prominent tumor signatures is the binding of IGF-I to its receptor in the PI3K/AKT pathway. Once activated, PI3K phosphorylates and activates the serine/threonine-specific protein kinase AKT. AKT activation mediates processes that increase cellular growth, cell survival, cell motility, and angiogenesis, as well as the inhibition of apoptosis. Accordingly, many tumor cells exhibit a resistance to apoptosis that lends them a significant survival advantage compared to non-tumor cells [9]. Moreover, angiogenesis is important to the growth of many tumors. For instance, important GFs that affect angiogenesis include the VEGF family and EGF. Blocking them from binding to tumor cells can cause tumor-cell death [9,17,18,19]. Thus, interfering with the binding of hormones and GFs to tumor cells may represent an important strategy for combating tumor development and progression, respectively.

Taken together, the binding of hormones and GFs can have direct effects on metabolic processes in target cells. Considering that most tumors develop from the clonal expansion of one cell mutated via somatic mutation [5,20], a process that is strongly influenced by the binding of signaling molecules, it is clear that key steps of tumor development and progression are driven by hormones and GFs [5,9,15]. The following section will describe how hormones and GFs influence the stages of tumor growth and progression.

### 1.2. Hormones and GFs in Tumor Growth and Progression

Tumor growth and progression represent a cascade of different developmental stages that are influenced by the interplay between a variety of factors in the tumor’s intra- and extracellular environment. The process of tumor progression can be divided into several distinct steps: (1) somatic mutation, (2) clonal expansion, (3) intraluminal cell proliferation and intraluminal lesion formation, (4) invasion, (5) dissemination, (6) the formation of micrometastases, (7) resistant tumor clones, (8) angiogenesis, and ultimately (9) metastases [5] (Figure 1). These stages of tumor growth and progression are substantially affected by the hormone and GF signaling [5,9,13,15].

Upon the initial somatic mutation and transformation of a healthy cell into a tumor cell, processes of cell growth and expansion are initiated. Tumor cells can mediate their own growth by the ectopic secretion of hormones and GFs [6]. In addition, despite their ability to perform ectopic secretion, tumor cells are frequently characterized by an overexpression of GF receptors, which contributes to their increased responsiveness to GFs. For instance, the EGF Receptor (EGFR) is commonly overexpressed in head or neck cancer cells, and ErbB-2/HER2 is overexpressed in breast cancer cells [5,21]. Moreover, signaling factors involved in various mitogenic pathways can be mutated in different tumor cells, which typically increase tumor growth. For example, RAS is mutated in 25% of human cancers [5].

Somatic mutation is followed by the clonal expansion of the tumor cell, which is frequently supported by GFs such as EGF and IGF-I [22]. Clonal expansion often contributes to intraluminal lesions in the organ [23]. Intraluminal cell proliferation and the formation of intraluminal lesions are frequently associated with abnormal hormonal and GF signaling. For example, ErbB-2/HER2 is commonly overexpressed in 40% of ductal carcinoma in situ (DCIS) breast cancer cases [24]. Intraluminal lesions occur upon the disruption of the basal cellular membrane, which represents the basis for tumor progression. The formation of intraluminal lesions is followed by the invasion of neighboring tissues and organs by tumor cells [25,26]. Invasion is possible due to acquired increased motility, loss of epithelial polarity, protease secretion, and the epithelial-mesenchymal transition (EMT) of tumor cells. Features of tumor-cell invasiveness are controlled by a multitude of hormones and GFs. For example, the activation of the Estrogen Receptors (ERs) by estrogens and Progesterone Receptors (PRs) by progesterone in transformed cells directly affects the cellular proliferation status, intracellular adhesion, cell motility, and morphology [14] via the activation of multiple transcription factors that alter the expression of genes involved in these processes. Moreover, signaling factors like the TGF-β or autocrine stimulatory loop signaling via EGFR have been demonstrated to promote tumor invasiveness [27,28] by causing a shift in cell adhesion properties of tumor cells, mainly through the downregulation of protease inhibitors, upregulation of protease secretion, or by causing a general increase in cell motility [5,28,29,30].

Tumor invasion is enhanced due to the EMT of tumor cells. The EMT is characterized by a mesenchymal stem cell-like stage of tumor cells, where initially polarized cells are transformed into non-polarized cells with increased motility that secrete ECM components [5,31]. A hallmark of cells that have undergone EMT is the loss of E-cadherin due to its mutational inactivation, transcriptional repression, or proteolysis [5]. E-cadherin is a key regulator of cell adhesion and the maintenance of an epithelial-cell phenotype [32]. In tumor cells, GFs like TGF-β, EGF, or Notch proteins regulate the expression of E-cadherin repressors [5,32,33] to maintain their stem cell-like phenotype.

Following invasion, tumor dissemination can take place. Dissemination is described as the process of tumor-cells spreading into other regions, or organs of the body via the entry (extravasation) and exit (intravasation) through the lymphatic, capillary, or blood vessel system [5,34,35]. Tumor dissemination is significantly influenced by signaling between tumor cells and surrounding cells such as endothelial cells and macrophages. For example, paracrine signaling between different cell types in the tumor microenvironment (TME), as well as the proinflammatory and hypoxic conditions during tumor progression, can facilitate the increased secretion of TGF-β by macrophages and ultimately aid tumor cells in entering the bloodstream [5,36]. Once in the vascular system, tumor cells become micrometastases or cells that spread from the primary tumor to surrounding tissues, or distant organs, which represent the main cause of tumor lethality [37,38]. When micrometastases invade another tissue or organ, they can start forming secondary tumors or metastases. The process of metastasis formation is supported by the fact that the disseminated tumor cells mainly represent cell death- and therapy-resistant clones of the primary tumor, giving them a substantial survival advantage compared to surrounding cells [9]. As previously mentioned, such cells have a significantly upregulated PI3K-AKT pathway that facilitates their resistance to apoptosis. The upregulation of the PI3K-AKT pathway represents a consequence of strong signaling from GFs such as IGF-I and EGF-like ligands [5,9]. Moreover, the secretion of factors such as Colony-Stimulating Factor 1 (CSF1) by tumor cells and surrounding macrophages, as well as NRGs, can significantly influence tumor growth and resistance [5,39].

Finally, to gain access to nutrients and oxygen, as well as to proliferate and survive, both primary tumors and metastases can facilitate their vascularization or angiogenesis [40]. De novo blood vessel formation is supported by the recruitment of mature endothelial cells and bone marrow-derived endothelial progenitor cells (EPCs) [25]. GFs such as VEGFs, FGFs, and TGF-β play crucial roles in this process. Moreover, the secretion of proangiogenic GFs by EPCs is important in mediating the angiogenic switch, which is when the antagonism between proangiogenic and antiangiogenic factors switches towards a proangiogenic result [5,41,42]. One of the hallmarks of cancer progression is hypoxia, which contributes to the plasticity and heterogeneity of tumors. Most solid tumors contain a hypoxic core, due to rapid cellular growth that outgrows the blood supply. In these circumstances, hypoxia-inducible factors (HIFs) play a central role in the adaptation of tumor cells to their new environment, dramatically reshaping their transcriptional profile. HIF signaling is modulated by a variety of factors including hormones and GFs which activate signaling pathways that enhance tumor growth and metastatic potential and impair the response to therapeutic agents.

In this review, we will describe the importance of hormones and GFs during cancer onset and progression with a particular focus on hypoxia and the interplay with HIF proteins. We will also discuss how hypoxia influences the efficacy of cancer immunotherapy. Indeed, hypoxia is considered a determinant of the immune-excluded phenotype and a major hindrance to the success of adoptive cell therapies (ACTs).

## 2. Hypoxia-Inducible Factors

In addition to biological macromolecules such as hormones and GFs, physiological factors are also intimately linked to tumor progression by affecting the TME. One physiological factor that critically influences the TME is the level of oxygen tension (a.k.a. partial oxygen pressure). Under normal physiological conditions, the oxygen tension ranges from 8 to 100 mmHg [43], a state that is referred to as physioxia. When the demand for oxygen exceeds the environmental supply, tissues are described as being in a state of hypoxia. While hypoxia is a normal occurrence during development (e.g., during mammalian embryogenesis), it is also associated with medical conditions [44,45,46,47,48]. Hypoxic areas are present in most solid tumors because of cellular proliferation outgrowing the blood supply [49,50,51], and solid tumors need to become angiogenic to grow beyond 1–2 mm in diameter [52,53,54,55]. Transient acute hypoxia and chronic hypoxia lead to a metabolically heterogeneous TME [56,57,58,59,60,61,62], which creates a strong selective pressure on cells, favoring the growth of more aggressive tumor clones. Therefore, hypoxia is generally clinically associated with poor prognosis across multiple tumor types and is also one of the main causes of therapeutic resistance [59,60,63,64,65,66].

Cellular homeostasis is maintained under hypoxic conditions by a family of heterodimeric transcription factors known as hypoxia-inducible factors (HIFs). HIF proteins contain alpha (HIF-α) and beta (HIF-β or ARNT) subunits and exist as three isoforms (HIF-1, HIF-2, and HIF-3). HIF-1α and HIF-2α are structurally similar except for their transactivation domains. HIF-1α generally binds hypoxia response elements (HREs) close to gene promoters, while HIF-2α targets transcriptional enhancers [67,68,69,70,71,72]. This may explain why they have both common and unique target genes despite binding identical HREs. While HIF-1 plays a major role in the regulation of genes involved in glycolysis, HIF-2 is mainly involved in pluripotent stem cell maintenance and angiogenesis, enhancing the pro-tumorigenic phenotype [73,74,75,76,77]. Finally, HIF-3α lacks a transactivation domain, suggesting that it may suppress the other HIF isoforms [78,79,80,81].

### 2.1. HIF-Dependent Regulatory Mechanisms

HIF proteins bind to canonical DNA sequences known as HREs, activating the expression of genes involved in a plethora of cellular processes [67,82,83,84,85,86,87,88,89,90,91,92,93]. While HIF proteins clearly exert their functions in a transcription-dependent manner, HIF proteins also impact important cellular processes in manners that are independent of transcription. For example, HIF-1α has been shown to repress DNA replication and induce cell cycle arrest in response to hypoxia independent of transcriptional regulation [94]. Thus, HIFs are the main drivers of cancer progression, exerting their functions through transcription-dependent and -independent mechanisms.

### 2.2. Oxygen-Dependent Regulation of HIF Signaling

As HIF signaling pathways can have significant impacts on cell function, the production and activity of HIF proteins are tightly regulated. HIF signaling is regulated through numerous cellular mechanisms. Some mechanisms of regulating HIF signaling are oxygen-dependent (e.g., responsive to changes in oxygen tension), while others are oxygen-independent.

HIF activity is largely regulated in an oxygen-dependent manner, such that HIF is active under hypoxia but inactivated in response to normal oxygen tensions. Under physioxia, HIF-1α is produced but is rapidly degraded, resulting in minimal levels of detectable HIF-1α protein. HIF-1α degradation is mediated by prolyl hydrolases (PHDs) which hydroxylate two proline residues within the oxygen-dependent degradation domain of HIF-1α [95,96,97,98] (Figure 2). Hydroxylated HIF-1α is polyubiquitinated by the von Hippel-Lindau tumor suppressor (pVHL), leading to HIF-1α degradation by the 26S proteasome [95,96,97,98]. The factor inhibiting HIF (FIH) protein provides an added layer of HIF repression, suppressing HIF-1 transcription in an oxygen-dependent manner by preventing the recruitment of co-activators [99]. In contrast, under hypoxic conditions, PHDs are inhibited which stabilizes HIF-1α, permitting its nuclear translocation and dimerization with constitutively active HIF-1β. This HIF-1α/β heterodimer then binds to HREs to activate transcription of hypoxia-responsive genes including those involved in promoting glycolytic metabolism, angiogenesis, and survival [67,68,83,100,101,102,103]. Abnormal HIF-1α and consequent upregulation of its target genes have been shown to occur in a wide range of solid tumors as they progress to malignancy [104].

HIF signaling is also regulated in oxygen-independent manners through mechanisms including transition metals (TMs), reactive oxygen species (ROS), reactive nitrogen species (RNS), and mechanical forces. In the late 1980s, a study discovered that TMs could induce hypoxia-like conditions, demonstrating increased erythropoietin gene expression following exposure to TMs such as Co(II), Ni(II), and Mn(II) [105]. Follow-up studies investigating the mechanisms underlying the regulation of erythropoietin gene expression led to the discovery of HIF-1 [68]. It was later found that exposure of cells to various TMs such as Ni(II), Co(II) [106,107], As(III), Cr(VI), and V(V) [108,109] stabilize HIF-1α. This is thought to occur through the inhibition of HIF-1α hydroxylation via two independent mechanisms: (1) substitution of iron by metal ions or (2) iron oxidation in the hydrolases [110,111]. Recent studies provide convincing support for the second hypothesized mechanism. Kaczmarek et al. detected HIF-1α stabilization in human lung epithelial cells in response to exposure to metal and metalloid ions, including some that cannot be used to substitute for iron in the hydroxylases [111]. Additional papers supported that HIF-1α can be stabilized by metal anions that are not iron substitutes (e.g., As(III), Cr(VI), and V(V)) [108,109].

In addition to TMs, reactive molecules such as ROS and RNS have also been shown to stabilize HIFs in a manner that is oxygen-independent. ROS have been discovered to promote HIF-1α transcription and translation in both hypoxia and normoxia [112,113]. For example, superoxide (SO) and hydrogen peroxide have been implicated in ROS-mediated HIF-1α stabilization [113,114,115,116]. Mechanistically, SO has been shown to inhibit VHL from binding to HIF-1α and thus inhibiting PHD activity [115,117] but the mechanisms underlying hydrogen peroxide-mediated HIF stabilization remain unknown. With regard to RNS, increased nitric oxide (NO) leads to HIF-1α accumulation and enhanced DNA binding activity in normoxic conditions [118,119]. This is thought to be mediated by HIF-1α post-translational modifications such as S-nitrosylation [120,121] or by inhibiting PHD activity [122]. The ability of ROS and RNS to mimic hypoxia in this manner suggests that ROS/RNS and hypoxia use similar mechanisms to stabilize HIF-1α [123]. The effects of ROS and RNS on HIF stability have been extensively and elegantly reviewed by Movafagh et al. [124].

Another mechanism triggering HIF stabilization independent of changes in oxygen levels is mechanical force. For example, VEGF-A expression can be induced under non-hypoxic conditions in response to the mechanical stress of the myocardium [125]. A study examining mechanical stress on rat hearts revealed stress-induced VEGF-A induction is preceded by HIF-1α accumulation in the nuclei of cardiac myocytes in a manner that depends on stretch-activated channels and the PI3K/Akt/FRAP pathway that is known to induce HIF-1α [126]. Similar studies revealed that HIF-1α is induced upon the stretching of rat vascular smooth muscle cells [127], and HIF-1α and HIF-2α are induced in skeletal muscle microvascular endothelial cells upon stretching of rat muscles, suggesting that both HIF-1α and HIF-2α contribute to stretch-induced capillary growth under non-hypoxic conditions [128].

Taken together, these findings suggest that HIF signaling is regulated by a variety of oxygen-dependent and -independent mechanisms. Several additional mechanisms are emerging as key regulators of HIF signaling including regulation of HIF by GFs and hormones. In the next section, we will describe the mechanisms through which hormones and GFs impact HIF signaling.

## 3. Modulation of HIF Signaling by Hormones and GFs

Cancer cell proliferation is stimulated by a variety of hormones such as steroid hormones (SHs) as well as GFs. Likewise, SH and GF axes can also synergistically activate intracellular pathways that are involved in tumor initiation, progression, survival, and resistance to chemotherapy [129,130,131,132,133,134].

Estrogens are SHs that regulate the reproductive process, bone density, and brain function, as well as the endocrine, cardiovascular, and metabolic systems [129,135]. However, these steroids are also considered a risk factor for many types of malignancies such as breast, endometrial, ovarian, prostate, and thyroid tumors [136]. Estrogens mediate numerous biological processes mainly by binding to cognate intracellular receptors, namely Erα and Erβ, which function as ligand-activated transcription factors that induce the transcription of various target genes. In addition, acting through membrane-located Ers or additional estrogen-binding proteins, estrogens elicit rapid responses such as an increase in calcium and nitric oxide levels and the activation of certain kinases [136,137,138,139]. In this regard, several studies have suggested that the G-protein estrogen receptor (GPER) may mediate rapid estrogen actions in both normal and tumor contexts [140,141]. In relation to the pro-tumorigenic properties of estrogens, Erα activates genes involved in the promotion of cell proliferation and cell cycle progression [142,143], inhibition of apoptosis [144], stimulation of angiogenesis [145,146,147], migration and invasion [148], and EMT [149,150]. Beyond Erα, estrogen-induced GPER activation initiates intracellular signaling pathways and triggers changes in gene expression via heterotrimeric G proteins, thus impacting cancer growth, invasion, and metastasis [141,151,152]. Interestingly, GPER can also regulate the expression of pro-tumorigenic mediators such as inflammatory cytokines and angiogenic factors within the TME [153,154,155,156,157]. Of note, Erα and GPER mediate tumor-promoting responses to estrogens by engaging in functional interactions with GF tyrosine kinase receptors including EGFR, IGF-IR, and Fibroblast GF Receptor (FGFR) [132,158,159,160,161,162].

GFR activation represents one of the most common events in the development of human cancers, as GFR activity is implicated in tumor cell proliferation, survival, transformation, angiogenesis, and metastasis [163,164,165,166,167]. EGF, IGFs, FGF, and platelet-derived growth factor (PDGF) are some of the most well-studied GFs that activate numerous pro-carcinogenic signaling pathways including ERK/MAPK, JNK, PI3K/AKT/mTOR, STAT, and PKC transduction cascades [164,166,167,168]. Remarkably, GF/GFRs are attractive therapeutic targets in human cancers due to their hyperactivation and frequent overexpression in cancer cells [164,165,169,170]. Intriguingly, hypoxic mediators as HIFs have been recognized as important drivers of GFR expression and signaling through both oxygen-dependent and -independent mechanisms [171,172,173,174,175,176]. Indeed, high HIF-1α levels have been associated with the overexpression of both GFs and GFRs in human cancers [177,178].

### 3.1. Hormone-Dependent Regulation of HIF Expression and Signaling

HIF-mediated SH function is involved in non-hypoxic pathways that enable cancer cells to adapt to altered TMEs [179,180,181,182]. Previous studies have focused on the capacity of estrogen to regulate HIF-1α expression in diverse types of cancer cells including thyroid, ovarian, and breast cancer cells, mainly via the PI3K/AKT signaling cascade [180,183,184]. Accordingly, select ER modulators downregulate VEGF-induced angiogenesis by suppressing HIF-1α/VEGFR2 signaling as well as the activation of AKT and ERK axes in breast cancer cells [185]. Furthermore, estrogens downregulate HIF-2α mRNA and protein levels in ER-positive breast cancer cells [186]. These findings are in line with immunohistochemical data showing that high HIF-2α levels are associated with a better overall survival rate in patients with invasive breast tumors [187]. In addition, estrogens cooperate with hypoxia to regulate genes involved in tumor cell growth and differentiation, angiogenesis, protein transport, metabolism, and apoptosis [188]. For instance, estrogen/Erα and hypoxia/HIF-1 transduction pathways can converge on the modulation of the epigenetic modifier histone demethylase KDM4B [188,189,190], affecting cell cycle progression and growth in ER-positive breast cancer [189,191]. The functional interaction between Erα and HIF-1α transduction pathways in breast cancer has been corroborated by evidence showing that HIF-1α may confer resistance to ER antagonists and may be considered as a transcriptional target of Erα [192]. In this respect, it has been reported that Erα binds to estrogen-responsive elements (EREs) located within the HIF-1α gene, thereby enhancing HIF-1α transcriptional activation [192]. The importance of these findings is highlighted by clinical studies revealing that the expression of HIF-1α or a hypoxia metagene signature in ERα-positive breast tumors is associated with aggressive phenotypes and a poor response to endocrine treatment [193,194,195].

The effects of estrogen may be mediated by additional factors, many of which can synergize with and/or regulate HIFs to promote aggressive neoplastic features. For instance, the ERβ variant ERβ2, which is overexpressed in many tumors and associated with decreased overall survival, was shown to stabilize HIF-1α protein levels and promote a hypoxic gene signature in prostate cancer cells and proliferative and invasive phenotypes in triple negative breast cancer cells [196,197]. Moreover, Estrogen-Related Receptor Alpha (ERRα) can interact with HIF-1α, thereby inhibiting its ubiquitination and degradation to promote the adaptation of prostate cancer cells to hypoxic conditions [198]. Further supporting a relationship between estrogens and HIF-1α, our previous studies have demonstrated that estrogenic GPER signaling triggers VEGF expression by upregulating HIF-1α in normoxic breast cancer cells, cancer-associated fibroblasts (CAFs), and mouse xenografts of breast cancer, leading to neoangiogenesis and enhanced tumor growth [155].

### 3.2. GF-Mediated Regulation of HIF Expression and Signaling

A variety of GFs can activate HIF-1α/VEGF signaling in cancer cells. For instance, it has been shown that under normoxic conditions, activation of the EGFR pathway may lead to increased expression levels and transcriptional activity of HIF-1α via the PI3K/AKT cascade [199]. These findings are in line with data suggesting that increased PI3K activity can promote HIF-1α overexpression in several types of human cancers [65]. EGF was shown to induce the expression of CXC Chemokine Receptor 4 (CXCR4) and promote migration in normoxic non-small cell lung cancer (NSCLC) cells via HIF-1α and the PI3K/AKT and mTOR pathways [200]. Oxygen-independent regulation of HIF-1α through PI3K/AKT and rapamycin has also been shown to occur in breast cancer cells upon treatment with the ErbB3/ErbB4 ligand heregulin, which stimulates HIF-1α synthesis and VEGF-A expression [174].

In breast cancer cells, HIF-1α is required for upregulation and secretion of the pro-tumorigenic cytokine Stem Cell Factor (SCF) by EGF [201]. Likewise, HIF-1α-dependent phosphorylation of STAT3 by EGF stimulates proliferative and invasive responses in colorectal cancer cells [202]. Similarly, EGF-induced expression of the anti-apoptotic protein surviving occurs through HIF-1α, promoting resistance to apoptotic stimuli in breast cancer cells [203].

In addition to HIF-1α regulation by the EGF/EGFR transduction pathway in a hypoxia-independent manner, HIF proteins can regulate their own production through autocrine and additive loops by which hypoxia induces EGF/EGFR expression while the ensuing EGFR signaling synergizes with hypoxia to induce HIF-1α and/or HIF-2α to promote cancer cell survival, migration, and invasion [199,200,204,205,206]. For instance, hypoxia-induced EGFR activation promotes a migratory phenotype in head and neck squamous cell carcinoma (HNSCC) through both HIF-2α and its target gene TGF-α, which may act as an EGFR ligand [207].

Considering that one of the major effects of the EGFR/HIF loop is the induction of VEGF-A in both normoxic and hypoxic conditions [174,208,209], it is not surprising that EGFR-targeting agents exhibit antiangiogenic activity. Among these agents, the monoclonal antibody cetuximab (C225) decreased HIF-1α levels and VEGF-A production in in vitro and in vivo tumor models [210,211,212]. Of note, these effects contributed to vascular normalization and apoptosis-dependent regression of established tumors [212]. In line with these findings, overexpression of HIF-1α conferred resistance to cetuximab-dependent apoptosis and inhibition of VEGF-A secretion in several cancer cells, whereas HIF-1α silencing restored the sensitivity to cetuximab, leading to antitumor responses [213]. Moreover, downregulation of HIF-1α in cancer cells was required for cetuximab-mediated autophagy as well as the sensitivity of HNSCC cells to radiation [214,215]. Similarly, gefitinib and erlotinib reduced vessel formation, decreased vascular permeability, and improved tumor oxygenation in xenograft models through the inhibition of HIF-1α and the production of angiogenic factors such as VEGF-A and IL-8 by tumor cells [216,217,218]. These findings have important implications in cancer therapies employing EGFR inhibitors since resistance to EGFR-directed therapies can be rescued by simultaneously targeting HIF-1α or VEGF-A (reviewed in [209]).

Several lines of evidence suggest that HIF protein accumulation and its activity can occur in a variety of cancer cells in response to IGF-I stimulation [175,219,220,221]. For instance, the exposure of colon carcinoma cells to IGF-I induced HIF-1α protein expression and upregulated VEGF mRNA levels in a PI3K- and MAPK-dependent manner [175]. Accordingly, the involvement of MAPK signaling in the regulation of HIF-1α by IGF-I was observed in breast cancer cells using MEK1/2 inhibitors [222]. Moreover, the engagement of PI3K/AKT and MAPK by IGF-I signaling stimulated HIF-1α and its target genes GPER and VEGF-A in breast cancer cells and CAFs and promoted tumor angiogenesis [223]. In line with these findings, inhibition of the HIF-1α/VEGF axis by both synthetic and natural chemicals such as bisphosphonates, farnesyltransferase inhibitors, epigallocatechin-3-gallate, and dauricine suppressed IGF-I-induced angiogenesis in breast cancer, lung cancer, and HNSCC cells [219,224,225,226]. Of note, in Kaposi sarcoma cells, IFG-I-mediated induction of HIF-1α, HIF-2α, and VEGF-A was lessened by inhibiting IGF-IR, leading to decreased tumor vascularization [220]. Considering that hypoxia promotes HIF-1α-mediated transcription and secretion of IGF-II [227], it is perhaps unsurprising that HIF-1α inhibitors decrease IGF-II signaling and trigger anti-migratory effects in breast cancer cells [228]. These findings might provide new perspectives to clinical studies since IGF-IR inhibitors are non-therapeutic when used alone in cancer patients [229]. Considering that compensatory signals mediated by insulin receptor-A or IGF-IR/IR hybrids may contribute to the lack of efficacy of anti-IGF agents [229], simultaneous inhibition of the IGF system and HIF-1α might prevent compensatory signaling, therefore providing an alternative strategy for the therapeutic management of diverse tumors including breast cancer.

In addition to the EGF and IGF systems, PDGF and basic FGF (bFGF) act as survival factors in cancer cells by engaging stimulatory signaling mechanisms mediated by HIF-1α. For instance, bFGF triggers HIF-1α activation and VEGF-A release via the PI3K/AKT and MEK transduction pathways in normoxic breast cancer cells [176,230,231]. PDGF-BB has been shown to upregulate the Bcl-2 family member named Myeloid Cell Leukemia-1 (Mcl-1) by forming a transcription complex between β-catenin and HIF-1α in prostate cancer cells [232]. Blocking the PDGF system by imatinib inhibits HIF-1α and IGF-I expression in prostate cancer cells and xenograft models [233]. Moreover, HIF-1α promoted lymphatic metastasis of hypoxic breast cancer cells by activating PDGF-B transcription [234], while an AKT/HIF-1α/PDGF-BB autocrine loop mediates hypoxia-induced chemoresistance in liver cancer cells [235].

Taken together, these findings suggest that HIF stabilization and/or activation may be a prerequisite for changes in gene expression and activation of signaling pathways triggered by SHs and GFs (Table 1). These events may promote tumor growth and spread as well as altered responses to therapeutic agents. The next section will discuss the relationship between hypoxia and the immune-excluded phenotype.

## 4. The Contribution of Hypoxia to the Immune-Excluded Phenotype

During the last decade, significant advances were made in relation to cancer immunotherapy, and it is currently considered a promising therapeutic strategy for many types of cancer. However, despite the proven efficacy of immunotherapy treatments in treating hematological malignancies, solid tumors remain a challenge to treat [236,237,238,239,240]. A main challenge to overcome is the infiltration of T-cells in the tumor core, which is often hypoxic. Several mechanisms have been suggested to explain the immune-excluded phenotype; however, an integrated understanding of the role played by various determinants of immune exclusion is still lacking. Hypoxia is a hallmark of most solid tumors and one of the most relevant factors involved in the immune excluded phenotype [49,50,51,241]. Hypoxia is responsible for shaping the TME in a unique way, affecting both gene transcription and chromatin remodeling in cancer and immune cells and therefore plays a role in the formation of mechanical and functional barriers. Indeed, hypoxia is a major suppressor of the immune system that alters the expression of cytokines, inducing the expression of co-inhibitory ligands and recruiting immune-suppressive cell populations [241,242,243,244,245].

### 4.1. Physical Barriers

Cancer cells produce pro-angiogenic factors through upregulation of HIF-induced genes (Table 2), which are responsible for the spatial-temporal heterogeneity in blood flow observed in solid tumors [246,247]. Indeed, the new blood vessels are often abnormal in that they are leaky and have an aberrant structure, which further enhances the hypoxic conditions [55,248,249,250]. Consequently, reduced blood flow, acidosis, and increased interstitial fluid pressure affect the fitness and infiltration of T-cells in the tumor core, which leads to a poor clinical response to immunotherapy [243,251,252]. For example, Jayaprakash et al. (2018) showed that in preclinical prostate cancer, hypoxic zones resisted T-cell infiltration even in the context of Cytotoxic T-Lymphocyte-Associated Protein 4 (CTLA-4) and Programmed Cell Death Protein 1 (PD-1) blockade. However, treating these tumors with the hypoxia-activated prodrug TH-302 in combination with checkpoint blockade cured more than 80% of tumors in a mouse model [243].

HIF-1 upregulates VEGF-A, which is a main pro-angiogenic factor that is produced by fibroblasts and endothelial cells [292,293]. VEGF is also responsible for ECM fibrosis in cancer as it induces the secretion of fibronectin and collagen type-I by stromal cells, activated resident fibroblasts, and attracted fibroblasts [357,358]. Hypoxia-dependent increases in tumor stroma density are a common occurrence in cancer and are due to the increased secretion of fibrous material and collagen-modifying enzymes [253,254,255,256,359]. Depletion of HIF-1, but not HIF-2, inhibited collagen deposition in vitro and decreased stromal density in orthotopic tumor [257]. However, collagen degradation in certain regions of the TME has also been reported, due to the activity of matrix metalloproteinases (MMPs) [360,361]. Collagen degradation is an important step during TME remodeling and HIF-1 has been associated with transcriptional upregulation of MMP2 and MMP9 in vitro [258,362]. The interplay of these processes, especially the increased stromal stiffness in tumors, constitutes an important barrier against T-cell accessibility. This is caused by mechanical constraints and decreased bioavailability of signaling. Kuczek et al. (2019) cultured T-cells and human breast cancer cells in 3D conditions with high or low collagen density. While the proliferation of cancer cells was unaffected in high collagen density matrices, T-cell proliferation was significantly decreased, along with a higher ratio of CD4^+^ to CD8^+^ cells and downregulation of cytotoxic markers [363]. Recruitment and activation of fibroblasts are also mediated by a variety of cytokines (Table 2) including TGF-β that is secreted by cancer cells and other stroma cells. TGF-β transforms fibroblasts into CAFs, which are responsible for the secretion of fibrous material and cytokines [259,364,365]. CAFs play various roles in the TME, they are morphologically different from fibroblasts, and they possess an enhanced migratory potential and proliferative capacity. In hypoxia, a positive feedback loop has been reported between TGF-β and HIF proteins [366]. Moreover, HIF-1 activates the TGF-β/Smad signaling cascade leading to increased collagen deposition in dermal fibroblasts [367].

CAFs promote functional barriers that contribute to angiogenesis, tumor growth, metastasis, immune suppression, and metabolic reprogramming through the secretion of chemokines such as CXCL12. CXCL12 is also involved in the EMT in various tumor types [368,369]. Hypoxia-induced cytokines secreted by cancer cells, myeloid cells, and mesenchymal cells are involved in the EMT (Table 2). Such cytokines include Tumor Necrosis Factor Alpha (TNF-α), TGF-β, Interleukin-1 (IL-1), IL-6, and IL-8 [272,273,274,275,276,370,371]. Moreover, it has been shown that EGF, hepatocyte growth factor (HGF), bFGF, and PDGF are also involved in the induction of transcription factors responsible for EMT progression [277,278,372,373]. EMT signatures are inversely correlated with T-cell infiltration in NSCLC, urothelial cancer and with resistance to PD-1 blockade in melanoma [279,374,375,376]. However, a few recent studies showed a positive correlation between T-cell infiltration and stromal EMT-related gene expression, in various malignancies [377,378]. Overall, the TME contains a variety of physical barriers that act synergistically with functional barriers to create an immunosuppressive environment and impede T-cell activity.

### 4.2. Functional Barriers

Functional barriers embody a class of impediments that impair T-cell activity despite their interaction with cancer cells and include metabolic barriers such as nutrient deprivation in the TME, danger sensing pathways, soluble factors, and cell-intrinsic signaling (Table 2). These determinants affect T-cell penetration and expansion in the tumor nests.

Hypoxic conditions lead to a shift from oxidative to glycolytic metabolism due to the upregulation of HIF-related genes. This phenomenon is called the Warburg effect and it is almost universal in cancers even when oxygen levels are comprised in a physiological range, despite also occurring physiologically [379,380,381]. HIF-1 regulates enzymes responsible for glucose transport (GLUTs) to maintain cellular ATP pools [299,300]. In solid tumors, GLUT upregulation is usually correlated with poor prognosis [382]. HIF-1 is also responsible for the upregulation of enzymes involved in glycolysis and pH regulation [309,383,384]. Enhanced glycolysis leads to increased amounts of lactate in the TME, which is responsible for the acidification of the ECM and subsequent decreases in T- and NK cell function and survival [385,386]. Furthermore, hypoxia induces carbonic anhydrases (Cas), Na^+^/H^+^ exchanger (NHE1), bicarbonate transporters (SLC4A4), and Indoleamine 2,3 Dioxygenase (IDO), which contribute to the acidification and tryptophan depletion in the TME [310,311,316,387]. In addition to tryptophan, the TME can be deprived of other amino acids essential for T-cell activity such as arginine [388,389]. HIF-1 is also responsible for GLS1 upregulation in cancer cells, which enhances anabolic metabolism through the hydrolyzation of glutamine into glutamate [317,390]. Decreases in the extracellular concentrations of glutamine and leucine inhibit the differentiation of Th1 and Th17 effector lymphocytes while maintaining regulatory T-cell differentiation [391,392]. Finally, elevated concentrations of extracellular potassium are common in solid tumors due to the downregulation of the potassium channel Kv1.3 [318,319]. Ionic imbalance, low pH, low glucose, and insufficient amino acid presence in the TME impair effector T-cell proliferation and cytokine production, leading to T-cell exhaustion.

HIF proteins are responsible for the upregulation and secretion of a variety of chemokines responsible for myeloid cell recruitment (Table 2). In hypoxia, immune-suppressive populations such as regulatory T-cells, myeloid-derived suppressor cells (MDSCs), and tumor-associated macrophages (TAMs) infiltrate the tumor site and create a functional barrier to the infiltration of T- and NK cells [393,394,395]. TAM receptor kinases such as Tyro3, Axl, and Mertk promote the phagocytosis of apoptotic cells, binding to “eat-me” signals which are displayed on apoptotic cell membranes [396]. Several studies have reported that TAM receptors are involved in the hypoxic response and play a variety of roles in cancer cells and tumor-infiltrating immune cells. Different signaling cascades may lead to the regulation of distinct TAM-associated functions [349,350,351]. Despite some evidence suggesting that TAM receptors may be involved in the immune-suppressive phenotype, their function requires further investigation. Moreover, HIF-1 induces the expression of CD47 which is a “don’t eat-me” signal (CD47/signal regulatory protein (SIRP)-α axis) that blocks prophagocytic signals and promotes tumor escape from immune surveillance [397,398,399,400]. Hypoxia also upregulates CD39 and CD73 which drive the shift from a proinflammatory to an anti-inflammatory environment mediated by an accumulation of adenosine in the TME [401,402]. Adenosine receptors on the surface of T-cells are upregulated by HIF-1 and HIF-2, and adenosine intake by T-cells leads to the accumulation of intracellular cyclic adenosine monophosphate (cAMP) [403,404]. cAMP plays an important role in immunosuppression, inhibiting T-cell receptor-induced T-cell activation [405,406,407]. Another functional barrier is the hypoxia-dependent downregulation of proteins necessary for immune cell recognition. For example, hypoxic tumors exhibit downregulation of major histocompatibility class-I (MHC-I) molecules, which impairs their recognition by cytotoxic T-cells (CTLs) [348]. In vivo investigations have revealed that hypoxia triggers the inhibition of IFN-γ–dependent MHC-I upregulation, a phenomenon that is reversible upon re-oxygenation [408].

Several comprehensive reviews have recently commented on the role of hypoxia as a determinant of the immune-excluded phenotype in solid tumors, and this chapter provided an overview of the major mechanisms involved. Immune-excluded tumors are different from homogeneously-infiltrated tumors, as they contain gradients of exclusion. Immune-excluded tumors may occur because of numerous underlying mechanisms and represent a significant challenge for immunotherapy. Gradients of immune infiltration are peculiar within each tumor environment and are likely not present in silent tumors. In cold tumors, a lack of chemo-attraction may trigger a predominant phenotype rather than the presence of barriers. The next section will provide some perspectives on the interplay between hypoxia and immunotherapies.

## 5. Perspectives on the Interplay between Hypoxia and Immunotherapies

Hypoxia is associated with a poor response to radiotherapy and chemotherapy and, therefore, is considered a negative prognostic indicator [409,410]. Hypoxia promotes tumor progression through angiogenesis, stemness, increased cancer cell survival, and metastasis. In the last few years, hypoxia-activated prodrugs (HAPs) have been tested in clinical trials as strategies to reduce hypoxic areas and improve cancer therapies. HAPs are inactive compounds that are converted to active drugs in hypoxic tissues [411,412]. However, low partial oxygen pressure is a common feature of many healthy tissues, for example in secondary lymphoid organs where the oxygen level is approximately 2.5%. Immune cells require HIF-mediated responses for their effector functions and their inhibition can significantly impair their functionality. Class I HAPs require mild hypoxia for their activation, necessitating the discovery of HAPs that require more severe hypoxia (Class II) to avoid undesired side effects [413]. Clinical trials using HAPs alone reported disappointing results, and studies assessing the efficacy of HAPs combined with immunotherapy in the treatment of solid tumors are currently underway [414,415,416]. Jayaprakash et al. (2018) reported that hypoxic zones were prevalent in preclinical prostate cancer and resisted T-cell infiltration, even in the context of CTLA-4 and PD-1 blockade [243]. Indeed, HIF-1 directly binds the promoter of PD-L1, inducing its transcription in various tumor cells [353,417]. In clear cell renal cell carcinoma, HIF-2 triggers PD-L1 upregulation [418,419]. The PTEN/PI3K pathways are responsible for HIF-1-mediated upregulation of CTLA-4, PD-L1, and HLA-G checkpoint molecules in several different mice and human tumor cell lines. Jayaprakash et al. (2018) showed that combination therapy with TH-302 HAP and T-cell immune checkpoint blockades (CTLA-4 and PD-1) cured more than 80% of tumors in a mouse prostate-derived TRAMP-C2 model. It is thought that the combination of the two treatments increased T-cell infiltration into hypoxic areas due to improved vascularization. Moreover, both myeloid-derived suppressor cells and granulocytic subsets were reduced in the TME. However, the results were less conclusive in spontaneous TRAMP models than in the ectopic models [243].

Approaches to improve the efficacy of immunotherapies in hypoxic tumors might include improving the function of the existing microvascular network. As previously mentioned, the ECM is often very dense in solid tumors and this may increase the pressure on the vasculature, leading to its collapse [420,421,422]. Another potential strategy may be the normalization of abnormal blood vessels produced in hypoxic conditions through metronomic dosing of VEGFs inhibitors [421,423,424]. Both approaches may be useful to increase oxygenation in the tissues and assist in chimeric antigen receptor (CAR) T-cell delivery. Two clinical trials (NCT03634332 and NCT02563548) are currently assessing the efficacy of a combination of PEGPH20 and pembrolizumab for patients with “Hyaluronan High” solid tumors. PEGPH20 is a pegylated recombinant human hyaluronidase that enzymatically degrades hyaluronan in the tumor stroma. Hyaluronan is a major component of the stroma, and the purpose of these clinical trials was to reduce the stromal density to improve the function of the vascular network and facilitate pembrolizumab penetration. However, a clinical trial (NCT01839487) reported that PEGPH20 in combination with nab-paclitaxel and gemcitabine did not provide any benefit to patients with Stage IV pancreatic ductal adenocarcinoma.

A major issue associated with ACT is the off-target toxicity, and recent studies indicated that hypoxia may be exploited to obtain more precise CAR T therapies. A hypoxia-inducible transcription amplification system (HiTAsystem) was developed to control CAR expression in T-cells (HiTA-CAR-T) [425]. An anti-Her2 CAR was expressed only in hypoxic environments, and HiTA-CAR T showed significant hypoxia-dependent tumor suppression in murine xenograft models. Moreover, no toxicity to livers expressing human target antigens in naïve mice was detected. Kosti et al. (2021) also developed a hypoxia-inducible CAR construct in which the transcription was mediated by HIF-1 (HypoxiCAR) [426]. The expression of a pan-ErbB-targeted CAR within hypoxic solid tumors demonstrated anti-tumor efficacy without off-tumor toxicity in murine xenografts models. Interestingly, they also reported that HypoxiCAR T-cells were not excluded from HIF-1-stabilized regions of the tumor (Figure 3).

During the last decade, several studies have reported that hypoxia may be beneficial for CAR T-cell activation, leading to increased T-cell effector function. Ectopic HIF expression has also been used to improve the antitumor efficacy of CD8^+^ T-cells and may be an interesting strategy to apply to ACT cancer therapy. For example, Gropper et al. (2017) showed that CTLs activated under hypoxic conditions (1% oxygen) have improved intrinsic cytotoxicity rather than migration potential inside tumors in vitro [427]. CTLs had increased amounts of granzyme-B in each granule, leading to improved mouse survival. However, hypoxia inhibited CD8^+^ T-cell proliferation in culture and increased exhaustion markers such as T-cell immunoglobulin, Mucin-Domain-Containing 3 (TIM3), and Lymphocyte Activation Gene 3 (LAG3). Conversely, CTLA4 and PD1 expression remained unaltered in hypoxia. Caldwell et al. (2001) performed similar experiments in CTLs at 2.5% oxygen and found an increase in effector functions and cell survival and a decrease in cytokine secretion and T-cell expansion [428]. Other reports indicated that 1% oxygen did not modify CTL cytotoxicity but induced IL-10 secretion and upregulation of CD137 and CD25 [429]. Roman et al. (2010) performed a similar study on CD4^+^ T-cells and found that stimulation under hypoxic conditions increased the secretion of effector cytokines, especially IFN-γ [430]. Deletion of the transcription factor Nuclear Factor Erythroid 2-Related Factor 2 (Nrf2) impaired the enhancing effect of hypoxia. Another study showed that HIF-2α, but not HIF-1α, drove broad transcriptional changes in CD8^+^ T-cells, resulting in increased cytotoxic differentiation and cytolytic function against tumors [431]. Moreover, a HIF-2 form that is insensitive to FIH was delivered with anti-CD19 CAR T-cells. These CAR T-cells showed enhanced cytolytic function in vitro and in a B-cell lymphoma xenograft ACT mouse model. Finally, genetic deletion of VHL and genes encoding for PHDs caused the accumulation of HIF-1 and HIF-2 in T-cells, enhancing their cytotoxic differentiation, and improving the rejection of primary and metastatic melanoma tumors [431].

Finally, Berahovich et al. (2019) characterized anti-CD19 and anti-BCMA human CAR T-cells generated at 18% and 1% oxygen [432]. Under hypoxia, CAR T-cells were associated with reduced proliferation and a less differentiated phenotype (higher CD4:CD8 ratio). Cytolytic activity and PD-1 upregulation was found to be similar in hypoxic and normoxic CAR T-cells. However, cytokine production and granzyme release were significantly decreased in hypoxia. It seems that oxygen tension may play a role during CAR T-cell differentiation, as has been previously reported [432]. The Berahovich et al. (2019) study found that hypoxia favored the differentiation in central memory T-cells (Tcm) compared to effector memory T-cells (Tem). The increase in central memory CAR T, and in less differentiated CAR T-cells in general, may be favorable and improve the efficacy of therapeutic treatments, as reported in previous clinical trials [432].

This observation regarding the prevalence of Tcm in hypoxia contrasts with findings reported by other studies, in which T-cell activation in hypoxia leads to increased differentiation of T-cells towards more lytic effector cells and impairs the proliferation of naïve and Tcm [428,429,433,434]. Indeed, Tem mainly rely on glycolysis instead of oxidative phosphorylation to support their metabolism, which is more favorable under hypoxic conditions. The inconsistencies reported across different studies may be explained by differences in the experimental protocols adopted, the type of stimulus used for activation, and the type of cells used. The timing of activation and analysis may also play a role, as HIF-1 is mainly stabilized during the early stages of hypoxia while HIF-2 is stabilized during the later stages and plays a main role in chronic hypoxia.

## 6. Conclusions

Hormones and GFs are emerging as key regulators of HIF signaling, with impacts on tumor growth, metastasis, and response to therapeutic agents. HIF proteins can thus be viewed as “central stations” that respond to various facets of the TME and facilitate the adaptation of tumor cells to their environment. Hormones and growth factors can shift the tumor toward a malignant phenotype, due to their influence on HIF proteins. Additional studies examining the connection between hormones, GFs, and hypoxia are vital, as the latter is a major determinant of the immune excluded phenotype. Furthering our understanding of the relationship between hypoxia and immune exclusion will indeed provide key insight into the development of novel and effective ACTs.

## Figures and Tables

**Figure 1 cancers-14-00539-f001:**
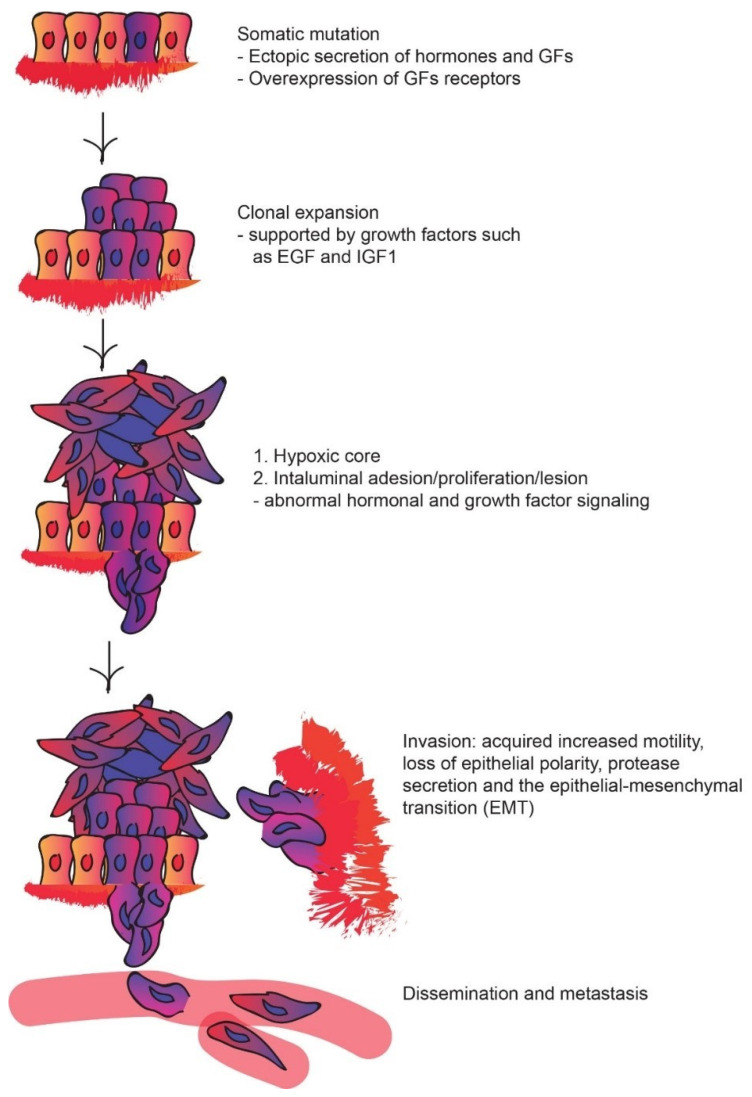
Schematic of tumor growth and progression (affected by hormone and GF signaling): somatic mutation, clonal expansion, intraluminal cell proliferation and intraluminal lesion formation, invasion, dissemination, the formation of micrometastases, resistant tumor clones, angiogenesis, and ultimately metastases.

**Figure 2 cancers-14-00539-f002:**
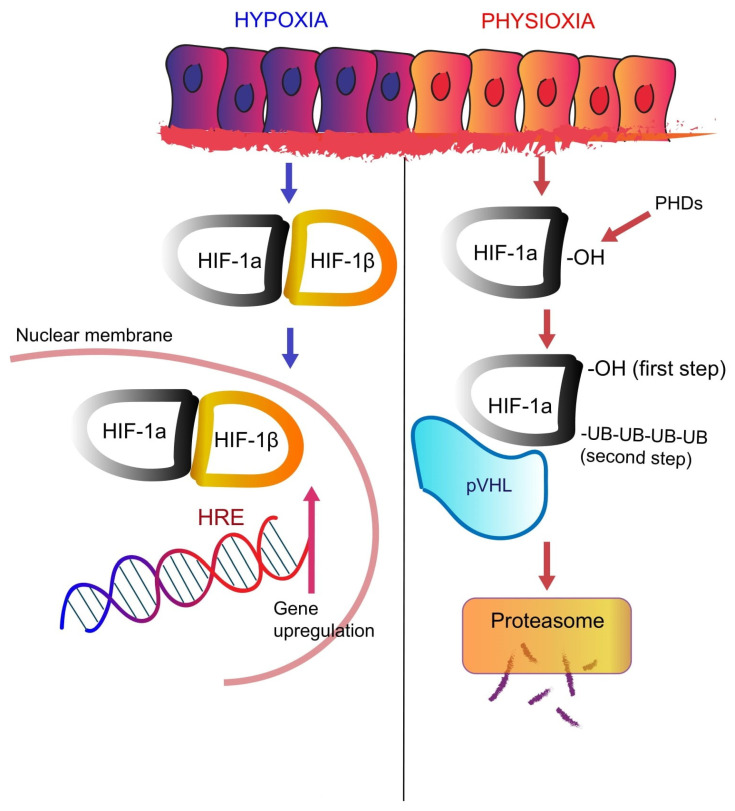
In physioxia (**right**), hypoxia-inducible factor (HIF) prolyl hydroxylase domain enzymes (PHDs) regulate the stability of HIF proteins by post-translational hydroxylation of two conserved prolyl residues in its α-subunit in an oxygen-dependent manner. Hydroxylation of HIF creates a binding site for pVHL that directs the polyubiquitylation of HIF-1α and its proteasomal degradation. In hypoxic conditions (**left**), HIF-1α binds to HIF-1β to form a heterodimer that acts as transcription factor, upregulating a variety of genes.

**Figure 3 cancers-14-00539-f003:**
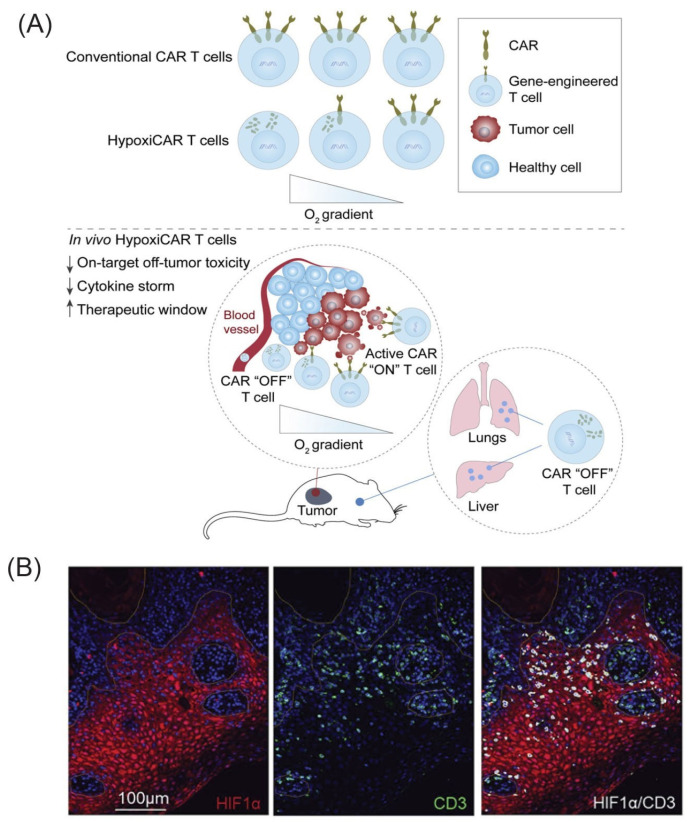
(**A**) Schematic of HypoxiCAR T-cell activation: A hypoxia-inducible CAR construct, whose transcription is mediated by HIF-1. (↑ means increase, ↓ means decrease) (**B**) HypoxiCAR T-cells are not excluded from HIF-1-stabilized regions of the tumor. Immunofluorescence images from a human oral tongue carcinoma. Nuclei are stained with DAPI (blue), anti-CD3 antibody (green), and anti-HIF-1α antibody (red); white denotes CD3 and HIF-1α co-localization. Scale bar 100 μm. (Reprinted with permission from Elsevier. Copyright (2021) Cell Reports Medicine [426]).

**Table 1 cancers-14-00539-t001:** Hormone and GF-dependent regulation of HIF expression/signaling (↑is increase, ↓is decrease).

Signaling	Cancer	Response	Reference
ERα-mediated estrogen signaling	Ovarian	↑ HIF-1α protein/signaling	[180]
	Thyroid	↑ HIF-1α protein	[183]
	Breast	↑ HIF-1α protein/signaling	[184]
	Breast	↓ HIF-2α mRNA/protein	[186]
	Breast	↑ HIF-1α mRNA	[192]
GPER-mediated estrogen signaling	Breast, CAFs	↑ HIF-1α mRNA/protein/signaling	[155]
EGF/EGFR signaling	Prostate	↑ HIF-1α protein/signaling	[199]
	NSCLC	↑ HIF-1α signaling	[200]
	Breast	↑ HIF-1α protein/signaling	[201,203]
	Colorectal	↑ HIF-1α mRNA/protein	[202]
Heregulin	Breast	↑ HIF-1α protein/signaling	[174]
IGF-I	Colon	↑ HIF-1α protein/signaling	[175]
	NSCLC, HNSCC	↑ HIF-1α protein/signaling	[219]
	Kaposi sarcoma	↑ HIF-1α and HIF-2α protein/signaling	[220]
	Breast	↑ HIF-1α and HIF-2α protein/signaling	[221,222,223]
bFGF	Breast	↑ HIF-1α protein/signaling	[176,230,231]

**Table 2 cancers-14-00539-t002:** Determinants of immune exclusion that are influenced by hypoxia.

Physical Barriers	Impediments to Direct Contact Between T-Cells and Cancer Cells	Reference
Stromal fibrosis	Epidermal growth factor (EGF), platelet-derived growth factor (PDGF), fibroblast growth factor 2 (FGF2), CXCL12, TGF-β, zinc finger E-box binding homeobox 1 and 2 (ZEB1, ZEB2) proteins, Snail, Slug, Twist, Goosecoid, FOXC2, LOX, PLOD1, PLOD2, P4HA1, P4HA2, MMP2, and MMP9	[253,254,255,256,257,258,259,260,261,262,263,264,265,266,267,268,269,270,271]
Epithelial-mesenchymal transition (EMT)	Tumor necrosis factor α (TNF-α), TGF-β, interleukin 1 (IL-1), interleukin-6 (IL-6) and interleukin-8 (IL-8), hepatocyte growth factor (HGF), basic fibroblast growth factor (bFGF), epidermal growth factor (EGF), platelet-derived growth factor (PDGF)	[272,273,274,275,276,277,278,279,280,281,282,283,284,285,286,287,288,289,290,291]
Vascular access	VEGF-family, angiopoietin-2 (Ang-2), transforming growth factor beta (TGF-β), platelet-derived growth factor B (PDGFB), placental growth factor (PGF), connective tissue growth factor (CTGF), stem cell factor (SCF), stromal cell-derived factor 1 (CXCL12), leptin, endoglin, nitric oxide synthase 2, haemoxygenase-1, endothelin-1 (ET-1), VEGF receptor-2, endothelial receptor tyrosine kinase (Tie-2)	[83,234,292,293,294,295,296,297,298]
Functional Barriers	Biological or metabolic interactions between cancer, stromal and immune cells limiting migration, function, and/or survival of T-cells	
Metabolic barriers	Warburg effect: enzymes glucose transporters (GLUTs 1–3), pyruvate dehydrogenase kinase 1 (PDK1), lactic dehydrogenase A (LDHA) and pyruvate kinase M2 subtype (PKM2), mono-carboxylate transporters (MCTs)	[299,300,301,302,303,304,305,306,307,308]
	Acidification of the TME: carbonic anhydrases (CAs), Na^+^/H^+^ exchanger (NHE1), bicarbonate transporters (SLC4A4)	[90,309,310,311,312,313,314,315]
	Amino acids depletion and ionic misbalance: indoleamine 2,3 dioxygenase (IDO), Glutaminase 1 (GLS1), Kv1.3	[316,317,318,319,320,321,322,323,324,325,326]
Soluble factors and “Don’t eat-me” signals	Myeloid cells recruitment and activity: CCL5, CXCL12, CXCR4, VEGF, Sema3A, CCL28, endothelin 1 and 2, TGF-β	[327,328,329,330,331,332,333,334]
	Adenosine signaling: CD39, CD73	[335,336]
	“Don’t eat me” signals: CD47/signal regulatory protein (SIRP)-α axis	[337,338]
Tumor cell-intrinsic signaling	Pathways involved in immune escape: extended PI3K pathway signaling, β-catenin/signaling, STAT-3 activation, MAPK signaling, p53 signaling	[199,339,340,341,342,343,344,345,346,347]
	Downregulation of molecules necessary for effector immune cells recognition: major histocompatibility class-I (MHC-I)	[348]
	TAM receptor tyrosine kinases: Tyro3, Axl and Mertk	[349,350,351,352]
Dynamic barriers	Interactions between cancer and T-cells resulting in limited function	
	Checkpoint/ligand interactions: upregulation of CTLA-4, PD-L1, HLA-G	[353,354,355,356]

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
