# Peer review of "Multifaceted Interplay between Hormones, Growth Factors and Hypoxia in the Tumor Microenvironment"

_cancers, 2022, doi:10.3390/cancers14030539_

Round 1

Reviewer 1 Report

The current manuscript reviewed the multifaceted interplay between hormones and growth factors and hypoxia in tumor microenvironment. This is a very important area in tumor biology, and also important to anti-cancer drug development. Hypoxia, as a key feature of tumor progression, plays critical roles for tumor progression, aggressiveness and also resistance to chemotherapy. Hypoxia-inducible factors (HIFs) interact with different proteins, altering the signal pathways. It is helpful for biological studies and development of anti-cancer drugs targeting related proteins, so the current manuscript will impact for the community who are interesting for HIFs and related hormone receptors and growth receptors.

The manuscript was well organized and written, except the following a few suggestions:

  1. on page 7, HIF-1alpha should be correct format, not HIF1-alpha
  2. on page 8, part 3 of the manuscript, the authors set only one subtitle, 3.1. Hormone and GF-dependent regualtion......., it would be better if they could further divide them into several parts using proper subtitles to indicate the focus. That will be helpful for the readers.
  3. Another suggestion for part 3 of the manuscript is to use table or chart to summarize the relationships of hormone and GF-dependent regulations with HIFs.

Author Response

1. on page 7, HIF-1alpha should be correct format, not HIF1-alpha

Thank you for your comment. We corrected the format of HIF-1alpha on page 7 and checked all the others throughout the whole manuscript.

2. on page 8, part 3 of the manuscript, the authors set only one subtitle, 3.1. Hormone and GF-dependent regulation......., it would be better if they could further divide them into several parts using proper subtitles to indicate the focus. That will be helpful for the readers.

Thank you for your comment. We further divided part 3 into several parts using proper subtitles.

3. Another suggestion for part 3 of the manuscript is to use table or chart to summarize the relationships of hormone and GF-dependent regulations with HIFs.

Thank you for your comment. We added Table 1 to the manuscript, to summarize the relationships of hormone and GF-dependent regulations with HIFs.

Reviewer 2 Report

In this review Lappano et al. explore the correlation between hypoxia, secretion of growth factors and hormones and changes to the tumor immune microenvironment eventually leading to immune suppression that may impact patients’ response to immunotherapy. Overall, this work is complete and provides interesting insights on the effects of hypoxia on the tumor microenvironment through different mechanisms.

I have a few minor concerns:

  • The authors don’t seem to discriminate between VEGFs, I am under the impression that they are discussing VEGF-A but this should be stated more clearly;
  • The quality of the figures could be improved and there is a lack of homogeneity throughout the review. The authors may want to try using BioRender or similar softwares. 

Author Response

I have a few minor concerns:

  • The authors don’t seem to discriminate between VEGFs, I am under the impression that they are discussing VEGF-A but this should be stated more clearly;

Thank you for your comment. As suggested, we specified VEGF-A throughout the manuscript.

  • The quality of the figures could be improved and there is a lack of homogeneity throughout the review. The authors may want to try using BioRender or similar softwares. 

Thank you for your comment. We used Adobe Illustrator to create figures that are more homogeneous.